# Motion-Guided Prior Support and Polarity Interaction for Event Stream Super-Resolution

## Abstract

In this paper, we aim to enhance the representation of spatio-temporal semantics during the Event stream Super-resolution (ESR) reconstruction by leveraging inter-frame motion information. To this end, we propose a Motion-Guided Prior Support and Polarity Interaction Network (MPS-PI Net). The MPS-PI Net takes event frames as the primary input, while incorporating positive and negative event streams as auxiliary inputs. The MPS-PI Net contains two novel designs: Motion-Guided Semantic Prior (MGSP) Module and Bipolar Semantic Interaction and Fusion (B-SIF) Module. In the MGSP module, we capitalize on inter-frame optical flow information to seamlessly integrate semantic cues derived from previously reconstructed frames into the super-resolution reconstruction process of the current frame. This integration provides valuable prior support for reconstructing the content of the current frame with greater accuracy. Building upon the prior semantic information introduced by the MGSP module, within the B-SIF Module, we initially undertake self-representational enhancement for both positive and negative polarity semantics independently. Following this, we conduct an interactive fusion of these two polarity semantics to fully harness their unique advantages. Experimental results unequivocally demonstrate that our proposed MPS-PI Net achieves competitive performance on many ESR datasets.

## 1 Introduction

Event cameras, bio-inspired sensors Zhu et al. (2018), asynchronously capture pixel-level brightness changes, providing high temporal resolution, low latency, and wide dynamic range Brandli et al. (2014). These features enable applications in extreme conditions where conventional cameras struggle, such as high-speed motion and HDR scene analysis Bardow et al. (2016); Cannici et al. (2020); Kim et al. (2021). However, hardware constraints limit their spatial resolution, hindering performance in vision tasks. Recent research Duan et al. (2021); Huang et al. (2024); Liang et al. (2024b) has focused on Event Stream Super-Resolution (ESR), which faces unique challenges due to the sparse, asynchronous, and polarity-sensitive nature of event data compared to RGB-based methods. Current ESR methods Duan et al. (2021); Huang et al. (2024); Li et al. (2019; 2021); Liang et al. (2024b); Wang et al. (2020); Weng et al. (2022) can be categorized into super-resolution based on raw event streams and super-resolution based on event frames. First category of methods aimed to directly generate high-resolution event streams from low-resolution event streams inputs, typically by preserving the spatiotemporal distribution of events through Spiking Neural Networks (SNNs) Li et al. (2019; 2021) or leveraging auxiliary frame information Wang et al. (2020). However, these methods often require high-quality image frames as external guidance and incurring considerable memory overhead. The second category of ESR methods Duan et al. (2021); Huang et al. (2024); Liang et al. (2024b); Weng et al. (2022) reformulate raw event streams into dense, frame-like representations. Subsequently, these methods typically employ an RGB-based paradigm to perform super-resolution reconstruction on event frames. In Duan et al. (2021); Weng et al. (2022), positive and negative events—despite being inherently misaligned in space and time—are typically merged into unified representations. While such fusion helps suppress noise to some extent, it inevitably leads to partial cancellation of opposite polarities, thereby diminishing polarity-specific structural details. Moreover, treating positive and negative events as a single representation may overlook their distinct dynamic patterns, limiting the network's ability to fully exploit their complementary

characteristics. To this end, learning-based approaches have adopted polarity-separated processing Huang et al. (2024); Liang et al. (2024b), where positive and negative event streams are handled independently to better preserve polarity-specific dynamics and mitigate the interference between opposite polarities. Despite their success in event super-resolution, two issues remain. First, spatial-motion semantic consistency in event frame sequences is underexplored. Liang et al. (2024b) uses low-res semantics from prior frames, but this struggles to guide current frame reconstruction effectively, compromising spatiotemporal coherence. Given high temporal resolution, adjacent frames share similar semantics; thus, we leverage inter-frame motion and prior spatial semantics for reconstruction support. Second, the complementary pattern of semantic meanings between positive and negative polarity events is insufficient. Currently, Methods Huang et al. (2024); Liang et al. (2024b) merely conduct a direct fusion of the semantic meanings associated with both polarities. However, due to variations in light intensity changes and the differing mechanisms of information generation, the structured semantic meanings embedded in positive and negative polarity events are distinct. Consequently, we aim to enhance the semantic representations within each polarity separately and subsequently facilitate the interaction and fusion of semantic meanings between the positive and negative polarities.

Above all, we propose a Motion-Guided Prior Support and Polarity Interaction Network (MPS-PI Net), which uses event frames as primary input and positive/negative event streams as auxiliary inputs to leverage polarity-specific dynamics. MPS-PI Net contains two key modules: the Motion-Guided Semantic Prior (MGSP) Module, which utilizes inter-frame optical flow to incorporate semantic cues from prior frames for current frame reconstruction, enhancing frame coherence; and the Bipolar Semantic Interaction and Fusion (B-SIF) Module, which first enhances positive and negative polarity semantics separately, then fuses them interactively to exploit their strengths while minimizing interference. The main contributions of this work are as follows:

- We introduce the MGSP Module, aimed at bolstering the semantic coherence across frames subsequent to super-resolution processing.
- We devise the B-SIF Module to effectively tap into and leverage the complementary semantics inherent in positive and negative event streams.
- By integrating the MGSP Module and the B-SIF Module, we present the MPS-PI Net. Experimental findings confirm that our MPS-PI Net attains remarkable and competitive performance across a multitude of ESR datasets.

## 2 RELATED WORK

**Video Super-resolution.** Video Super-Resolution (VSR) recovers high-res videos from low-res ones by enhancing spatial resolution. BasicVSR Chan et al. (2021) uses bidirectional propagation for long-term video temporal modeling. BasicVSR++ Chan et al. (2022) builds upon BasicVSR by introducing several architectural improvements, including second-order deformable alignment, flow-guided feature propagation, and instance normalization. VRT Liang et al. (2024a) combines mutual attention and self-attention, which are respectively responsible for inter-frame alignment and intra-frame information preservation. PSRT Shi et al. (2022) proposes a progressive spatio-temporal recurrent Transformer framework for video super-resolution, which gradually refines features across both spatial and temporal dimensions. With the rapid advancement of diffusion models Ho et al. (2020) in image generation, recent studies have begun exploring their potential for video super-resolution. Upscale-A-Video Zhou et al. (2024) uses diffusion for real-world VSR with temporal consistency. SeeClear Tang et al. (2024) introduces semantic distillation, where high-level semantics guide pixel-level learning.

**Event Stream Super-resolution.** Event Stream Super-resolution is challenging due to event data's unique spatiotemporal traits. Early methods focused on preserving event streams' inherent properties. For instance, Li et al. Li et al. (2019) proposed the Event Count Map (ECM) to describe event spatial distribution using a non-homogeneous Poisson model, but struggled with accuracy under large upsampling factors. To address this, Wang et al. Wang et al. (2020) introduced GEF, which employs motion correlation probabilities and image guidance to improve spatial fidelity, though it degrades when image frames are unreliable. Subsequent efforts explored SNNs to better exploit temporal information. Li et al. Li et al. (2021) proposed a spatiotemporal constraint learning approach based on SNN dynamics, enabling joint modeling of spatial and temporal features. Others

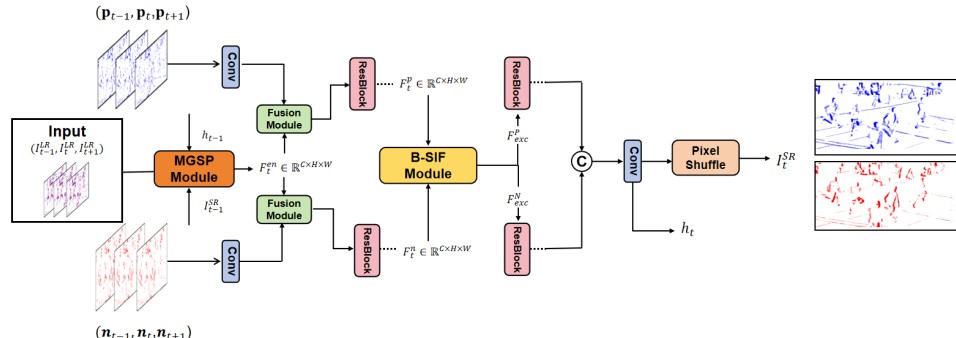

Figure 1: Overall architecture of our MPS-PI Net. MGSP Module represents the Motion-Guided Semantic Prior Module; B-SIF denotes Bipolar Semantic Interaction and Fusion Module.

transformed event streams into frame-like representations to leverage CNNs. Duan et al. Duan et al. (2021) introduced a 3D U-Net based on stacked event frames, while Weng et al. Weng et al. (2022) designed an RNN-based framework that combines temporal propagation and spatiotemporal fusion to effectively handle high-scale SR. More recently, methods such as BMCNet Huang et al. (2024) and RMFNet Liang et al. (2024b) have adopted polarity-separated branches to better preserve polarity-specific features and suppress mutual interference. However, they directly fuse cross-polarity features without boosting intra-polarity ones, limiting cross-polarity use and causing incomplete structure recovery. Unlike them, by leveraging inter-frame motion information, we further enhance the semantic representations of bipolar events, and subsequently strengthen the complementary advantages of polarity-specific semantics through the interaction between the two polarities.

## 3 METHOD

**Preparation of Event Frames.** For a set of event streams generated by event cameras, we can represent them as a sparse stream $\mathcal{E} = \{\mathbf{e}_k\}_{k=1}^{N}$, where $N$ represents the number of events, each event $\mathbf{e}_k \in \mathcal{E}$ can be represented by a tuple $(x_k, y_k, t_k, p_k)$, denoting spatial coordinates, timestamp and polarity respectively. Subsequently, we partition $\mathcal{E}$ into positive events $\{\mathbf{e}_k\}_{k=1}^{N_p}$ and negative events $\{\mathbf{e}_k\}_{k=1}^{N_n}$ based on their polarity $p_i = \pm 1$. We convert $\{\mathbf{e}_k\}_{k=1}^{N_p}$ and $\{\mathbf{e}_k\}_{k=1}^{N_n}$ into event frames Maqueda et al. (2018), which characterize the spatial distribution of events. Accordingly, two-channel event representations can be constructed from $\mathcal{E}$, consisting of positive event tensor $\mathbf{p}_t \in \mathbb{R}^{H \times W}$ and negative event tensor $\mathbf{n}_t \in \mathbb{R}^{H \times W}$.

### 3.1 OVERVIEW

An overview of our proposed Motion-Guided Prior Support and Polarity Interaction Network (MPS-PI Net) is depicted in Fig. 2. To better utilize the polarity-specific dynamics and enhance structural representation, we separate the positive and negative events for independent processing, and incorporate them as auxiliary inputs to the model. At each timestamp $t$, the low-resolution input $I_t^{LR} \in \mathbb{R}^{3 \times H \times W}$ is composed of three components: the positive event tensor $\mathbf{p}_t \in \mathbb{R}^{H \times W}$, the negative event tensor $\mathbf{n}_t \in \mathbb{R}^{H \times W}$, and the aggregated event frame $\mathbf{e}_t \in \mathbb{R}^{H \times W}$. To better exploit temporal continuity, we concatenate event representations from three consecutive timestamps $t-1, t$ and $t+1$ as the primary input to the model. The main input $(I_{t-1}^{LR}, I_t^{LR}, I_{t+1}^{LR})$, previous hidden state $h_{t-1}$ and output $I_{t-1}^{SR}$ are first processed by our proposed **Motion-Guided Semantic Prior (MGSP) Module**. This design enhances the temporal coherence and spatial consistency of event stream, resulting in enhanced features $F_t^{en} \in \mathbb{R}^{C \times H \times W}$ that serve as a more reliable foundation for subsequent reconstruction. The enhanced features $F_t^{en} \in \mathbb{R}^{C \times H \times W}$ are then integrated with the corresponding positive event tensors $\mathbf{p}_{t-1}, \mathbf{p}_t, \mathbf{p}_{t+1}$ and negative event tensors $\mathbf{n}_{t-1}, \mathbf{n}_t, \mathbf{n}_{t+1}$ via a Fusion Module. The fused features are subsequently processed by three stacked residual blocks to obtain higher-level representations $F_t^p, F_t^n \in \mathbb{R}^{C \times H \times W}$. Subsequently, the **Bipolar Semantic**

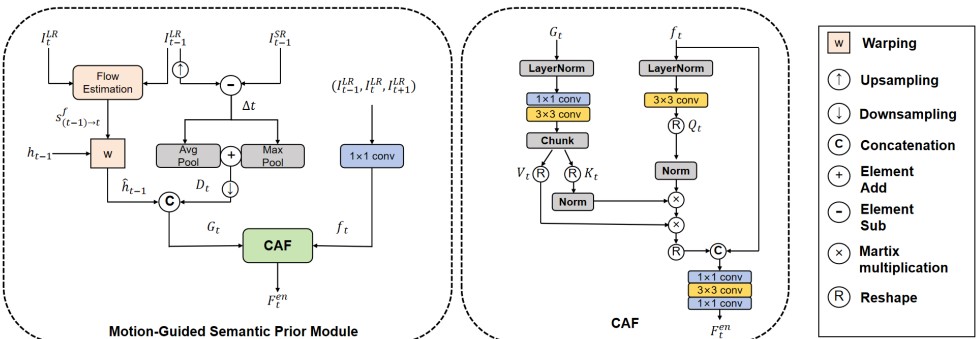

Figure 2: Architecture of the Motion-Guided Semantic Prior Module (MGSP). The MGSP Module is designed to incorporate semantic cues from previously reconstructed frames into the super-resolution reconstruction of the current frame.

**Interaction and Fusion (B-SIF) Module** is introduced to facilitate the interaction between positive and negative polarity, enabling the complementary features from each stream to be effectively leveraged for further refinement. Finally, the features from positive and negative branches are concatenated to generate the updated hidden state $h_t$, while the super-resolution event tensor $I_t^{SR}$ is produced through a pixel shuffle operation Shi et al. (2016).

## 3.2 MGSP MODULE

As described in Intro. 1, we propose the Motion-Guided Semantic Prior (MGSP) Module (Fig. 2). The MGSP Module is designed to incorporate semantic cues from previously reconstructed frames into the super-resolution reconstruction of the current frame, providing prior support for its content reconstruction. This module aims to reduce the reconstruction complexity of the current frame while enhancing the semantic coherence between frames after super-resolution processing.

Although our task is event-based super-resolution, the input event streams have been preprocessed into an RGB-like frame representation. This representation preserves the spatio-temporal information of events while being compatible with conventional optical flow estimators. Therefore, we employ a RGB-based optical flow network SPyNet (Ranjan & Black (2017)) to estimate the optical flow between the low-resolution input frames at time $t-1$ and $t$, capturing the pixel-wise motion across consecutive timesteps:

$$s^f_{(t-1)\to t} = \mathcal{F}_{\text{flow}}\left(I_{t-1}^{LR}, I_t^{LR}\right) \tag{1}$$

where $\mathcal{F}_{\text{flow}}$ denotes the SPyNet optical flow estimator, and $s^f_{(t-1)\to t} \in \mathbb{R}^{H \times W \times 2}$ represents the dense flow field indicating pixel-wise displacement from frame $t-1$ to $t$. Rather than using the estimated optical flow as an explicit input feature, we utilize it to temporally align the hidden state $h_{t-1}$. The hidden state $h_{t-1}$ contains rich temporal context and accumulated semantic information from previous frames, which is crucial for maintaining temporal consistency and guiding current predictions. However, directly using $h_{t-1}$ without considering inter-frame motion may lead to spatial misalignment, especially in the presence of fast-moving objects. To address this, we apply the estimated optical flow to warp $h_{t-1}$, obtaining an aligned hidden state $\hat{h}_{t-1}$ that is spatially consistent with the current frame:

$$\hat{h}_{t-1} = \mathcal{W}\left(h_{t-1}, s^f_{(t-1)\to t}\right) \tag{2}$$

where $\mathcal{W}(\cdot)$ denotes a warping operation based on bilinear sampling, which spatially transforms the previous hidden state $h_{t-1}$ according to the estimated flow field $s^f_{(t-1)\to t}$.

Secondly, to further provide prior reconstruction content for the super-resolution reconstruction of the current frame, we explicitly calculate the content of the super-resolved reconstruction of the previous frame by

$$\Delta_t = I_{t-1}^{SR} - \text{Upsample}(I_{t-1}^{LR}) \tag{3}$$

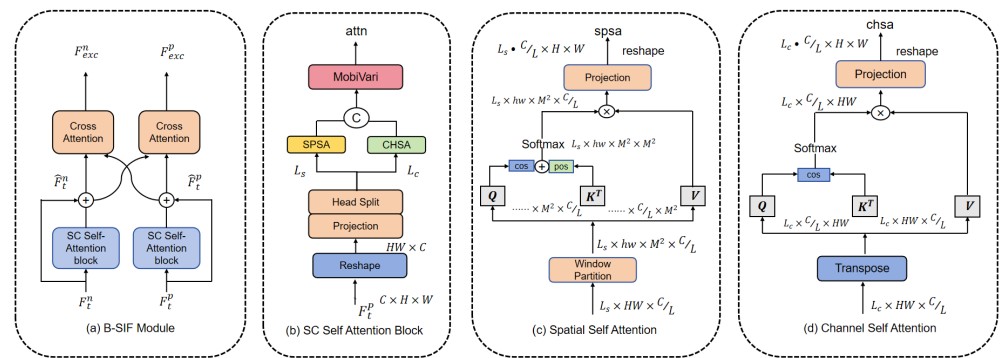

Figure 3: **(a)** Architecture of the proposed Bipolar Semantic Interaction and Fusion (B-SIF) module. **(b)** The parallel Spatial-Channel (SC) Self-Attention Block we designed. **(c)** Spatial Attention Mechanism. **(d)** Channel Attention Mechanism.

Here $\Delta_t$ represents the content of the super-resolved reconstruction of the previous frame. Besides, areas with larger values in $\Delta_t$ typically correspond to textures with higher complexity, stronger structural variation, or greater semantic importance, and are therefore considered critical for accurate restoration.

To utilize the reconstructed content of the previous frame as prior information for the super-resolution reconstruction of the current frame, we need to adjust the resolution of $\Delta_t$ to match that of $\hat{h}_{t-1}$. However, directly applying downsampling would result in significant loss of the reconstructed content from the previous frame. To address this, we employ both MaxPooling and AvgPooling to adjust the feature resolution, i.e.

$$D_t = MaxPool(\Delta_t) + AvgPool(\Delta_t) \tag{4}$$

The $MaxPool(\Delta_t)$ captures the most prominent regions, while the $AvgPool(\Delta_t)$ preserves smooth spatial semantics.

Finally, we concatenate the flow-aligned hidden state $\hat{h}_{t-1}$ and the content guidance map $D_t$ to form a unified guidance feature. However, directly concatenating this guidance feature with the raw input may lead to suboptimal fusion due to semantic mismatch and lack of feature selection. All channels are treated equally, allowing redundant or irrelevant information to propagate. To this, we adopt a Channel Attention Fusion (CAF) module Zhang & Yao (2024), where the current input feature $F_t$ extracted from the concatenation of $I_{t-1}^{LR}, I_t^{LR}, I_{t+1}^{LR}$, is used as Query $Q_t$, and the guidance feature serves as Key $K_t$ and Value $V_t$, i.e.

$$Q_t = \text{Conv}(\text{LayerNorm}(F_t)) \tag{5}$$

$$K_t, V_t = \text{Split}(\text{Conv}(\text{LayerNorm}([\hat{h}_{t-1}, D_t]))) \tag{6}$$

The attention weights are then computed by performing a scaled dot-product between $Q_t$ and $K_t$, followed by matrix multiplication with $V_t$:

$$F_{\text{att}} = A_t V_t, \quad A_t = \text{Softmax}(Q_t K_t^\top) \tag{7}$$

The $F_{\text{att}}$ is reshaped and concatenated with the original input $F_t$ followed by convolutional refinement to produce the final output feature $F_t^{en}$.

### 3.3 B-SIF MODULE

As shown in Fig. 2, after obtaining the enhanced feature $F_t^{en}$ from the MGSP Module, we incorporate it into the Fusion Module. The Fusion Module is designed to integrate the enriched temporal and structural prior cues provided by $F_t^{en}$ into the current polarity stream, enabling polarity-specific feature enhancement with clearer spatial semantics and more reliable temporal continuity. In this module, $F_t^{en}$ is separately combined with the positive and negative event streams, producing two

polarity-specific fused features. The fused features from the positive and negative branches are respectively passed through three stacked residual blocks, resulting in structurally enhanced and detail-preserving representations $F_t^p$, $F_t^n \in \mathbb{R}^{C \times H \times W}$ that retain key semantic content and suppress reconstruction artifacts. However, separately processing positive and negative event streams may overlook their complementary information, leading to incomplete representations. To address this, some excellent methods Huang et al. (2024); Liang et al. (2024b) attempted to fuse and exchange information from positive and negative streams after polarity separation. Nevertheless, they mainly focus on the interaction process between polarities, without fully enhancing the representation capacity of each stream before fusion. This limits the model's ability to capture rich intra-polarity semantics and cross-polarity relevance. Therefore, as shown in Fig.3a, we propose a Bipolar Semantic Interaction and Fusion (B-SIF) module that first enhances each polarity branch individually through parallel spatial-channel attention, and then performs cross-polarity exchange via bidirectional attention.

Specifically, the polarity-specific features $F_t^p$, $F_t^n$ are individually fed into the SC Self-Attention Block (Fig. 3b), which integrates both spatial and channel attention in parallel Choi et al. (2024). This block enables each polarity branch to capture long-range spatial dependencies and adaptively highlight informative channels, enhancing the intra-polarity representation. Taking the positive polarity stream as an example, the input feature $F_t^p \in \mathbb{R}^{C \times H \times W}$ is first split into $L$ heads. Among them, $L_s$ heads are allocated for Spatial Self-Attention (Fig. 3c), while the remaining $L_c$ heads are used for Channel Self-Attention (Fig. 3d), with $L_s + L_c = L$. The input feature $F_t^p$ is first reshaped and then split into two parts $F_{SPSA} \in \mathbb{R}^{L_s \times HW \times C/L}$ and $F_{CHSA} \in \mathbb{R}^{L_c \times HW \times C/L}$, which are respectively used as the input to the spatial self-attention (SPSA) and channel self-attention (CHSA) branches. The spatial self-attention is adapted from the window-based mechanism in SwinIR Liang et al. (2021), $F_{SPSA}$ is first partitioned into non-overlapping local windows of size $M \times M$, resulting in a feature representation of shape $\mathbb{R}^{L_s \times hw \times M^2 \times C/L}$, where $hw = HW/M^2$ is the number of windows per head, then three separate linear projections are applied to each windowed feature to generate the corresponding query $Q^s$, key $K^s$ and value $V^s$. And the channel self-attention builds upon the transposed attention structure introduced in Restormer Zamir et al. (2022), the input feature $F_{CHSA}$ is first transposed along the last two dimensions, resulting in a tensor of shape $\mathbb{R}^{L_c \times C/L \times HW}$, linear projections are then applied to this transposed feature to produce the query $Q^c$, key $K^c$, and value $V^c$. This dual-stream attention design facilitates the extraction of both local spatial features and global channel-wise dependencies in a lightweight and efficient manner. The attention computation is formulated as follows:

$$F_i^s = A_i^s \times V_i^s, A_i^s = \text{Softmax}\left(\cos(Q_i^s, (K_i^s)^T)/\tau + B\right) \tag{8}$$

$$F_j^c = A_j^c \times V_j^c, A_j^c = \text{Softmax}\left(\cos(Q_j^c, (K_j^c)^T)/\tau\right) \tag{9}$$

where $Q_i^s$, $K_i^s$, $V_i^s$ denote query, key, value for SPSA and $Q_j^c$, $K_j^c$, $V_j^c$ denote query, key, value for CHSA; $cos(\cdot)$ denotes cosine similarity; $\mathbf{B} \in \mathbb{R}^{M^2 \times M^2}$ is the relative positional bias, where $M$ denotes the window size; $\tau$ is a trainable scalar that is set larger than 0.01. Then we obtained SPSA and CHSA:

$$\text{SPSA} = \mathcal{P}_{sp}\left(\text{Concat}\left[F_1^s, \ldots, F_{L_s}^s\right]\right) \tag{10}$$

$$\text{CHSA} = \mathcal{P}_{ch}\left(\text{Concat}\left[F_{L_s+1}^c, \ldots, F_L^c\right]\right) \tag{11}$$

where $\mathcal{P}$ denotes reshape and projection layer. Finally, SPSA and CHSA are concatenated along the channel dimension and fused via the MobiVari network Sandler et al. (2018) to integrate attention features. A skip connection is then employed to produce the final output $\hat{F}_t^p$:

$$\hat{F}_t^p = \text{MobiVari}(\texttt{Concat}\left[SPSA, CHSA\right]) + F_t^p \tag{12}$$

After obtaining the two refined features $\hat{F}_t^p$, $\hat{F}_t^n$, we leverage the complementary information between positive and negative event streams to achieve more effective super-resolution reconstruction. Therefore, we introduce a Cross-Polarity Attention mechanism to facilitate the interaction between positive and negative event features. Taking the positive branch as an example, we reshape the positive feature $\hat{F}_t^p$ to obtain $V \in \mathbb{R}^{HW \times C}$, while the negative feature $\hat{F}_t^n$ is projected and reshaped to generate the corresponding query $Q, K \in \mathbb{R}^{HW \times C/R}$ where $R$ is the reduction ratio for channel compression. Therefore, the output of the positive feature, exchanged with negative feature, can be represented as:

$$F_{\text{exc}}^P = \text{Softmax}\left(QK^\top\right) V \tag{13}$$

| Methods | NFS-syn | | | RGB-syn | | EventNFS-real | | Param (M) | | | Inference time (ms) | | |
|---|---|---|---|---|---|---|---|---|---|---|---|---|---|
| | 2× | 4× | 8× | 2× | 4× | 2× | 4× | 2× | 4× | 8× | 2× | 4× | 8× |
| Bicubic | 0.558 | 0.479 | 0.495 | 0.1142 | 0.1324 | 0.760 | 0.899 | - | - | - | - | - | - |
| RSTT Geng et al. (2022) | 0.359 | 0.337 | 0.335 | 0.0827 | 0.0989 | 0.295 | 0.387 | 3.8 | 4.1 | 4.3 | 62.8 | 62.3 | 74.8 |
| RecEvSR Weng et al. (2022) | 0.386 | 0.343 | 0.327 | 0.0899 | 0.1075 | 0.355 | 0.431 | **1.8** | **1.8** | **1.8** | 13.7 | 19.4 | 19.9 |
| BMCNet Huang et al. (2024) | 0.293 | 0.280 | 0.281 | 0.0771 | 0.0926 | 0.257 | 0.364 | 2.6 | 2.7 | 3.1 | 17.4 | 23.6 | 38.9 |
| RMFNet Liang et al. (2024b) | 0.291 | 0.276 | 0.282 | 0.0783 | 0.0914 | 0.252 | 0.318 | 3.0 | 3.1 | 3.6 | **7.6** | **8.2** | **8.5** |
| Ours | **0.270** | **0.258** | **0.261** | **0.0758** | **0.0895** | **0.241** | **0.298** | 4.0 | 4.1 | 4.5 | 33.1 | 54.5 | 73.2 |

Table 1: Quantitative analysis comparison on real and synthetic datasets. Mean Squared Error ($MSE$) is used as the evaluation metric. Model Parameters (Param) and Inference time are calculated on the NFS-syn dataset. **Bold** and underline indicate the best and second-best results.

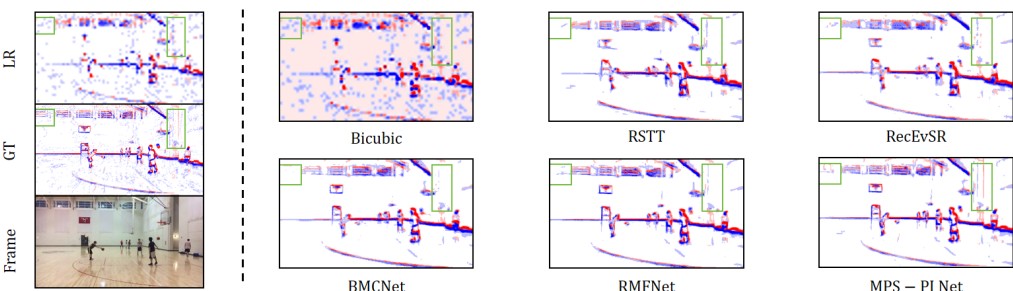

Figure 4: Qualitative analysis comparison on NFS-syn dataset. "GT" denotes the 4× ground-truth.

By obtaining the two exchanged features, complementary reconstruction and semantic alignment between positive and negative event streams are achieved, thereby enhancing the representational capacity and overall performance of super-resolution reconstruction.

**Training Objectives:** To maintain temporal coherence during training, following the approach in Weng et al. (2022), we split the event stream into overlapping segments of length T (where T=9 in our experiments), and then compute the mean square error for each segment using a sliding window, i.e. $\mathcal{L} = \sum_{t=1}^{T} \text{MSE}\left(I_t^{SR}, I_t^{HR}\right)$, $I_t^{SR}$ denotes the reconstructed event frame at timestamp $t$, and $I_t^{HR}$ is the ground truth.

## 4 EXPERIMENTS

**Datasets and Training Settings.** We validate our MPS-PI Net via experiments on real and synthetic event datasets. Public datasets with multi-scale LR-HR event pairs are rare due to challenges in aligning spatial-temporal info. The EventNFS dataset Duan et al. (2021) is the first real dataset to offer LR-HR pairs but is limited to low resolutions (min. 55×31) and small scaling factors (2×, 4×), with degraded quality at the lowest resolution. To enable higher-scale super-resolution, we convert high-frame-rate video datasets (NFS Kiani Galoogahi et al. (2017), RGB-DAVIS Wang et al. (2020)) into event streams using an event simulator Lin et al. (2022), creating synthetic NFS-syn and RGB-syn datasets. These provide high temporal resolution and are widely used in event-based vision, making them suitable for realistic event camera simulations.

For consistent benchmarking, we adopt the training configuration from Weng et al. (2022), using a batch size of 2, an initial learning rate of 0.001 with a decay factor of 0.95 every 4000 iterations, and a total of 100,000 training iterations. Mean Squared Error (MSE) is used as the evaluation metric, and all experiments are conducted on a Tesla V100 GPU.

### 4.1 COMPARISON WITH STATE-OF-THE-ART MODELS

We mainly compare MPS-PI Net with three learning-based methods:RecEvSR Weng et al. (2022), BMCNet Huang et al. (2024), and RMFNet Liang et al. (2024b). Other ESR methods rely on extra real-frame info or falter in complex scenes, hindering fair comparison. RecEVSR uses RNNs for

| Model | MGSP | B-SIF | | NFS-syn | EventNFS |
|-------|------|-------|------|---------|----------|
| | | SCSA | CA | | |
| model A | × | × | × | 0.288 | 0.334 |
| model B | ✓ | × | × | 0.273 | 0.320 |
| model C | ✓ | × | ✓ | 0.269 | 0.306 |
| model D | ✓ | ✓ | × | 0.265 | 0.304 |
| model E | ✓ | ✓ | ✓ | 0.258 | 0.298 |

Table 2: Ablation results for different components of our MPS-PI Net.

large upsampling. BMCNet and RMFNet split event streams by polarity, preserving traits and reducing interference. We retrain these models using their open code for fairness. Besides ESR baselines, we include Bicubic and RSTT from video/image super-resolution. We evaluate all methods on MSE, Model Params, and Inference Speed.

**Qualitative Analysis Results.** Fig. 4 shows 4× SR results on NFS-syn dataset (see **Appendix** for more). RGB-based methods like Bicubic and RSTT yield blurry edges and lose fine details on ESR tasks, likely due to the modality gap with event streams. RecEvSR recovers overall structure but struggles with details and sharp edges. BMCNet and RMFNet perform better by splitting event streams but still have detail reconstruction issues. Our MPS-PI Net uses two tailored modules to enhance structural and dynamic representations, effectively extracting complementary fine-grained features from both polarities for better detail and edge recovery.

**Quantitative Analysis Results.** As depicted in Table 1, our MPS-PI Net outperforms all datasets and magnification scales. Compared to the previous SOTA method RMFNet in ESR, our proposed method achieves an average MSE improvement 6.5% on NFS-syn and EventNFS-real datasets. Furthermore, compared to the video super-resolution method RSTT, MPS-PI Net achieves an average improvement of 22.3%. While MPS-PI Net introduces a modest increase in parameter count (4.0M–4.5M) and inference latency (33.1–73.2 ms), these values remain within a practical range for most deployment scenarios. The slightly higher computational cost is primarily due to our enhanced network design aimed at capturing complex spatiotemporal dynamics in event streams. Importantly, the consistent performance gains across synthetic and real-world datasets indicate that MPS-PI Net achieves a favorable trade-off between efficiency and reconstruction quality, making it well-suited for high-precision event-based vision tasks where accuracy is paramount.

## 4.2 ABLATION STUDY

**Validation of Main Components in MPS-PI Net.** To investigate the impact of main components in MPS-PI Net, we conducted a series of experiments and compared the 4× SR results on both synthetic and real datasets. The experimental results are shown in Table 2, we compared MPS-PI Net with four different variants with different settings. **MGSP** refers to the Motion-Guided Semantic Prior Module, while **B-SIF** denotes the Bipolar Interaction and Fusion Module. Additionally, **SCSA** and **CA** represent the SC Self Attention Block and Cross-Polarized Attention Block within the B-SIF, respectively. By comparing model A and model B, we observe that incorporating MGSP improves performance on both synthetic and real event datasets. This is because the MGSP leverages optical flow to utilize the reconstructed content from previous frames as prior knowledge for the super-resolution reconstruction of the current frame, thereby reducing the difficulty of super-resolving the current frame. Further comparisons between model B and model C, as well as between model B and model D, clearly demonstrate that SCSA and CA in B-SIF play distinct roles in enhancing the internal representations and facilitating stronger information exchange between the two streams, respectively. Specifically, SCSA improves the quality of internal representations, while CA enhances the efficient information exchange between the two streams, resulting in significant improvements in overall performance across both datasets.

**Analysis of MGSP.** We conducted a series of ablation studies to analyze the impact of each module in MGSP. Table 3 presents performance comparison with different configurations, where $s_{(t-1)\to t}^{f}$ stands for optical flow, $D_t$ represents content guide map, CAF refers to Channel Attention Fusion module. By comparing Exp1 and Exp2, we observed that incorporating optical flow provided richer

| Exps | $s^f_{(t-1)\to t}$ | $D_t$ | CAF | NFS-syn | EventNFS |
|------|------|------|------|---------|----------|
| Exp1 | × | × | × | 0.267 | 0.309 |
| Exp2 | ✓ | × | × | 0.265 | 0.304 |
| Exp3 | ✓ | ✓ | × | 0.259 | 0.300 |
| Exp4 | ✓ | ✓ | ✓ | 0.258 | 0.298 |

Table 3: Ablation study of MGSP components across various configurations

| Methods | Video Reconstruction | | | | | |
| | 2× | | 4× | | 8× | |
| | SSIM ↑ | LPIPS ↓ | SSIM ↑ | LPIPS ↓ | SSIM ↑ | LPIPS ↓ |
|---------|--------|---------|--------|---------|--------|---------|
| Bicubic | 0.431 | 0.314 | 0.448 | 0.476 | 0.423 | 0.528 |
| RSTT | 0.478 | 0.284 | 0.485 | 0.413 | 0.480 | 0.455 |
| RecEvSR | 0.469 | 0.299 | 0.474 | 0.420 | 0.466 | 0.471 |
| BMCNet | 0.492 | 0.258 | 0.494 | 0.381 | 0.489 | 0.427 |
| RMFNet | 0.490 | 0.251 | 0.487 | 0.385 | 0.486 | 0.423 |
| MPS-PI Net | **0.501** | **0.237** | **0.503** | **0.366** | **0.495** | **0.407** |

Table 4: Quantitative analysis results on downstream tasks of video reconstruction. Bold and underline indicate the best and the second-best performance.

motion information, resulting in superior reconstruction performance. Exp3 demonstrates that the integration of the content guidance map $D_t$ enhances performance on both synthetic and real-world datasets, which can be attributed to effectively incorporate reconstructed content from the previous frame as prior while preserving important structural details through the combined use of MaxPooling and AvgPooling. Furthermore, comparison between Exp3 and Exp4 shows that the CAF module effectively alleviates semantic mismatches and suppresses redundant or irrelevant information by performing adaptive channel-wise feature selection, thereby leading to better fusion and promotes final performance.

### 4.3 EVENT-BASED APPLICATION

**Video Reconstruction.** Video reconstruction is a fundamental task in event-based vision Rebecq et al. (2019); Stoffregen et al. (2020); Weng et al. (2021), and we have conducted a comparative analysis across various methods for this task. To begin, we applied each model for 2(4, 8)× super-resolution on the NFS-syn dataset, which had been downsampled by a factor of 16. Next, we employ a resampling method to reconstruct the event stream from the super-resolved event frames. And then, we utilized the E2VID Rebecq et al. (2019) for image reconstruction based on the super-resolved event stream. The quality of the reconstructed images was then assessed using structural similarity (SSIM) Wang et al. (2004) and perceptual similarity (LPIPS) Zhang et al. (2018) metrics. Table 4 summarizes the quantitative performance of event-based video reconstruction, showing that consistently outperform existing methods across multiple upscaling factors. Reconstructed images are provided in **Appendix**.

## 5 CONCLUSION

We presented a Motion-Guided Prior Support and Polarity Interaction Network (MPS-PI Net) for event stream super-resolution. The MPS-PI Net consists of two innovative components: Motion-Guided Semantic Prior (MGSP) Module and Bipolar Semantic Prior (B-SIF) Module. The MGSP Module utilizes inter-frame optical flow information to seamlessly integrate semantic cues from previously reconstructed frames into the current frame's super-resolution process. Meanwhile, the B-SIF Module directs the model to effectively harness the complementary semantics inherent in positive and negative event streams. Results from both real and synthetic datasets confirm our approach's competitive performance.

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

## A    APPENDIX

### A.1    DATASET AND TRAINING CONFIGURATION

**Detailed introduction of the dataset.** In the experiments, we utilized three datasets, including a real dataset EventNFS Duan et al. (2021), and two synthetic datasets NFS-syn Kiani Galoogahi et al. (2017) and RGB-syn Wang et al. (2020), generated using an event simulator Lin et al. (2022). We next provide a detailed description of the generation process for the two synthetic datasets. Initially, we downsampled the NFS dataset and the RGB-DAVIS dataset using Bicubic interpolation to generate low-resolution images. The original resolution of the NFS dataset is 1280 × 720, and it was downsampled by factors of 2×, 4×, 8×, and 16×. For the RGB-DAVIS dataset, whose original resolution is 1520 × 1440, we applied 2×, 4×, and 8× downsampling. Subsequently, event streams were synthesized from RGB videos using an event simulator, with default initial parameters applied during the conversion process. In the NFS-syn dataset, we defined the 16× downsampled data (80 × 45) as the minimum resolution and the 2× downsampled data (640 × 360) as the maximum resolution, thereby constructing LR-HR pairs at 2×, 4×, and 8× scales. For the RGB-syn dataset, the 8× downsampled data (190 × 180) served as the minimum resolution, and the 2× downsampled data (760 × 720) as the maximum resolution, resulting in LR-HR pairs at 2× and 4× scales. Finally, to enhance data diversity and robustness, we applied augmentation techniques such as random flipping and polarity inversion. The dataset was then randomly split into training and testing sets.

**Training Configuration.** During training, we applied several data augmentation techniques to the event stream data, including horizontal and vertical flipping of the event count images, as well as polarity inversion of the event streams. Each augmentation was applied with a probability of 50%. For all three datasets—EventNFS, NFS-syn, and RGB-syn—we randomly partitioned the data into training and testing subsets, and conducted training and evaluation separately on each dataset. The model was trained with a batch size of 2, an initial learning rate of 0.001, and a learning rate decay factor of 0.95 applied every 4,000 iterations. Training was conducted for a total of 100,000 iterations.

### A.2    MORE ABLATION STUDY

In the B-SIF module, we introduce a parallel spatial–channel self-attention mechanism, where the input features are split into L attention heads. In our implementation, L is set to 8. We systematically investigate how different allocations between spatial $L_s$ and channel $L_c$ attention affect performance through extensive experiments on NFS-syn dataset. Exp0 shows that using only channel attention results in a clear performance drop (0.265), likely because it captures global dependencies but fails to model the fine-grained spatial patterns critical in event stream. On the other hand, using only spatial attention (Exp4) also results in inferior performance in NFS-syn dataset. This is because relying solely on spatial attention, without a complementary channel attention mechanism, prevents the model from dynamically weighting the importance of different feature channels. As a result, it struggles to effectively select informative cues and suppress redundant or noisy features. In contrast, the combination of spatial and channel attention allows the model to jointly consider where and what to focus on, leading to more robust and discriminative representations. Among the configurations that combine both spatial and channel attention (Exp1–Exp3), we observe that performance improves as more attention heads are allocated to the spatial branch. Specifically, Exp3 ($L_s = 6, L_c = 2$) achieves the best performance, suggesting that spatial modeling plays a more critical role in our task. This can be attributed to the nature of event-based data, which exhibits strong spatial locality and structural sparsity. However, it is important to note that channel attention also plays a significant role, as it helps the model selectively focus on informative features across different channels, which is crucial for capturing the full complexity of event data. Allocating more capacity to spatial attention allows the model to better capture fine-grained spatial dependencies, which are essential for accurate reconstruction. In contrast, overemphasizing channel attention (e.g., Exp1) may lead to less effective spatial modeling, resulting in slightly degraded performance.

### A.3    MORE VISUAL RESULTS

**More Qualitative Comparison Results.** To further demonstrate the performance of MPS-PI Net, we present more 4× SR results of various methods on NFS-syn and EventNFS datasets, which is shown in Figure 5. It is obvious that our MST-PI Net demonstrates superior detail recovery

| Exps | $(L_s, L_c)$ | NFS-syn |
|------|------|------|
| Exp0 | (0, 8) | 0.265 |
| Exp1 | (2, 6) | 0.262 |
| Exp2 | (4, 4) | 0.261 |
| Exp3 | (6, 2) | **0.258** |
| Exp4 | (8, 0) | 0.262 |

Table 5: Ablation study on different head allocations between spatial and channel attention in the B-SIF module.

of the event streams, resulting in sharper edges. **Event-Based Video Reconstruction.** We performed 4× super-resolution on the downsampled NFS-syn dataset using several baseline and state-of-the-art methods, including Bicubic interpolation, RSTT Geng et al. (2022), RecEvSR Weng et al. (2022), BMCNet Huang et al. (2024), RMFNet Liang et al. (2024b), as well as our proposed MPS-PI Net. Subsequently, we employ a resampling method to reconstruct the event stream from the super-resolved event frames. The resulting super-resolved event streams were then passed through E2VID Rebecq et al. (2019) for image reconstruction. As shown in Figure 6, compared to existing methods, MPS-PI Net produce reconstructions with finer structural details and fewer visual artifacts, demonstrating the superiority and effectiveness of our approach.

## A.4 LIMITATION AND FUTURE WORK

Despite the strong performance of the proposed MPS-PI Net, it has limitations in effectively addressing noise in event streams. A key limitation of our current design lies in the insufficient handling of noise in event data. Compared with conventional RGB cameras, event cameras are more susceptible to noise caused by sudden illumination changes, sensor jitter, or background activity, especially in uncontrolled or low-light environments. Although our method incorporates stacked positive and negative event count maps along with polarity separation to suppress noise to some extent, it lacks an explicit denoising module. Future work could explore the integration of dedicated denoising mechanisms into the super-resolution framework to improve robustness and generalization in real-world scenarios.

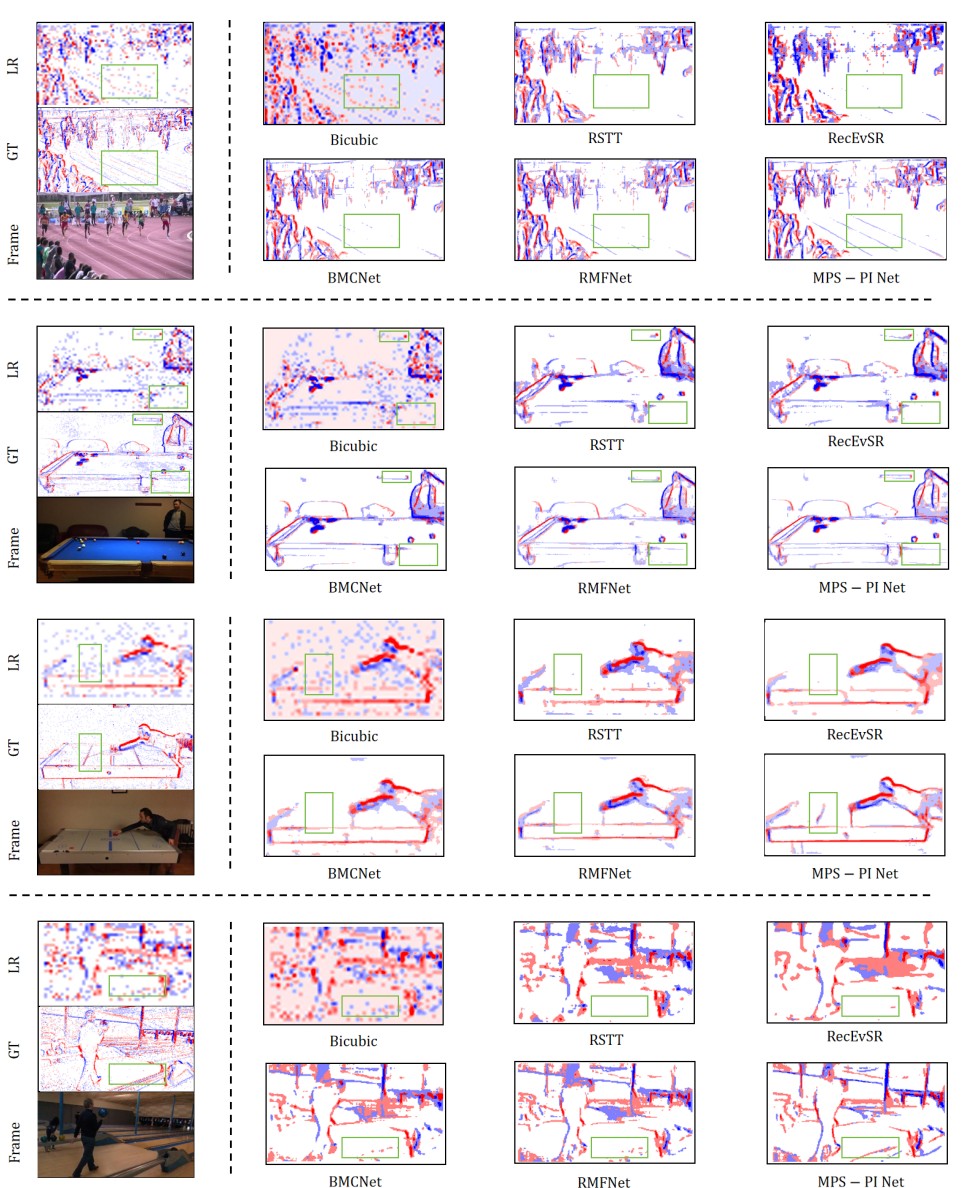

Figure 5: Qualitative Analysis Comparisons on the NFS-syn and EventNFS datasets. "GT" denotes the 4× ground truth. The top two rows show results on the NFS-syn dataset, while the bottom two rows correspond to the EventNFS dataset.

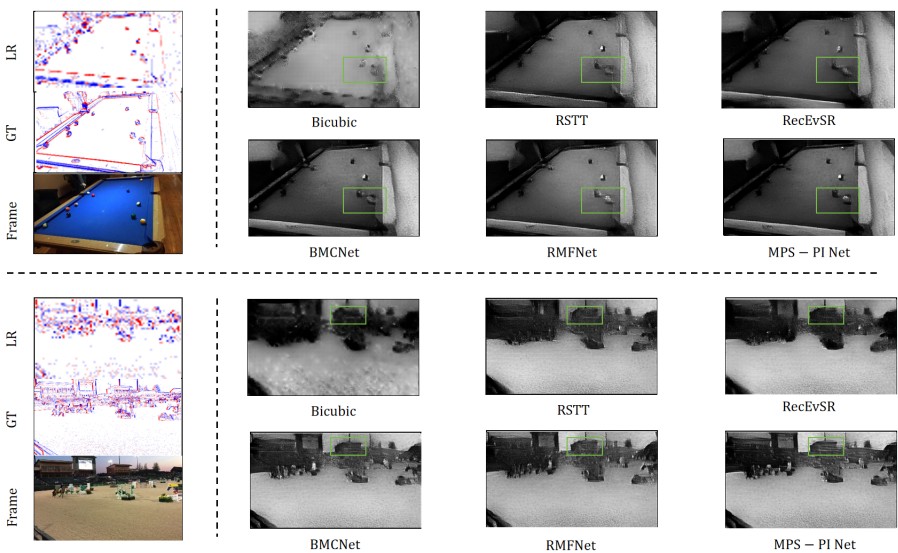

Figure 6: Qualitative Analysis Results for Event-based Video Reconstruction.

