# OpenReview forum: "Motion-Guided Prior Support and Polarity Interaction for Event Stream Super-Resolution"
_ICLR.cc/2026/Conference — ICLR 2026 Conference Withdrawn Submission_

### Official Review · Reviewer_35qN · 2025-10-27

**Soundness:** 2
**Presentation:** 2
**Contribution:** 2
**Rating:** 2
**Confidence:** 4

**Summary:**

This paper addresses the task of event stream super-resolution (ESR) and proposes a Motion-Guided Prior Support and Polarity Interaction Network (MPS-PI Net). The method consists of two main components: a Motion-Guided Semantic Prior (MGSP) module and a Bipolar Semantic Interaction and Fusion (B-SIF) module. Following prior ESR approaches such as BMCNet-ESR (CVPR’24), the authors evaluate their method on the same benchmark datasets and report state-of-the-art performance.

**Strengths:**

The proposed method demonstrates strong quantitative results, achieving state-of-the-art MSE scores across all datasets and upscaling factors presented in Table 1.

**Weaknesses:**

1.	**Unclear Motivation**: The motivation for the proposed method is not clearly articulated in the Introduction. The paper does not provide a convincing justification for why the MGSP and B-SIF modules are necessary, as the stated limitations of previous ESR methods are not thoroughly established.
2.	**Unrealistic Qualitative Examples**: The qualitative examples shown appear to be based on unrealistic inputs. The low-resolution (LR) event frames in Fig. 4 and Fig. 5 seem to have a far lower quality than what would be captured even by a standard event camera (e.g., a DAVIS346). This raises questions about whether the synthetic data is representative of a real-world scenario. It is suggested to conduct experiments on data more similar to real event streams. Additionally, the qualitative results on real data (from EventNFS) should be presented in the main paper, not relegated to the appendix (Fig. 5).
3.	**Insufficient Validation of Practical Utility**: The paper does not sufficiently validate the practical utility of the ESR task itself. The MSE improvements should be validated on downstream tasks, such as object detection or segmentation, to verify that the super-resolved event streams can indeed improve accuracy in real-world applications. While prior work (e.g., BMCNet) has included such validation, this paper lacks a similar analysis, making the significance of the MSE gains unclear.
4.	**Marginal Visual Improvements**: The visualized improvements against baselines in Fig. 4 and Fig. 6 appear marginal in terms of semantic content. While the method may produce slightly sharper or more numerous event edges, it is not immediately clear how this translates to a more meaningful or semantically rich representation of the scene. The practical benefit of these visual enhancements is not well-demonstrated.

**Questions:**

See weaknesses

---

### Official Review · Reviewer_P3ZD · 2025-10-27

**Soundness:** 2
**Presentation:** 2
**Contribution:** 2
**Rating:** 4
**Confidence:** 5

**Summary:**

This paper proposes MPS-PI Net, a method that leverages inter-frame motion information to enhance event-based super-resolution. The approach includes a Motion Guided Semantic Prior (MGSP) module and a Bipolar Semantic Interaction and Fusion (B-SIF) module. The MGSP module extracts motion and semantic information from previous frames to guide the super-resolution of the current frame, while the B-SIF module performs self-representation enhancement and fusion of positive and negative events. Experimental results demonstrate the effectiveness of these modules.

**Strengths:**

1. The proposed method achieves state-of-the-art performance on event-based super-resolution tasks while maintaining a relatively small number of parameters.

2. The method introduces two novel modules and effectively leverages motion information between event frames.

**Weaknesses:**

1. Limited novelty and contribution. The main contribution of this work lies in leveraging motion information between event frames to guide event-based super-resolution. However, similar approaches have been extensively explored in video-based super-resolution, and the authors do not provide targeted improvements that exploit the unique characteristics of event data. For example, event cameras have high temporal resolution, and the interval between event frames is much smaller than between video frames. Could motion information be extracted across multiple frames or adaptively, rather than simply using adjacent frames?

2. Potential lack of practical applicability. With the development of hardware, the resolution of event cameras has increased significantly—for example, the EVK4 reaches 1280×720 pixels. In contrast, this study focuses on super-resolution for very low-resolution inputs (55×31 and 80×45 pixels), which may not match real-world scenarios. It would be more practical to target high-resolution event cameras capturing small objects.

3. Missing important baselines. The proposed method uses a flow network for inter-frame motion estimation, which has been widely applied in video-based super-resolution [1,2]. The authors should clarify how their method differs from video-based approaches and include comparisons. Additionally, a key event-based super-resolution method, EventZoom, is not compared.

4. Unclear role of semantic information. From my understanding, the method primarily leverages temporal consistency and motion information between event frames, and the semantic component is not clearly demonstrated.

5. The line 65 “the structured semantic meanings embedded in positive and negative polarity events are distinct” is unclear. Positive and negative events typically occur together, reflecting object motion and shape, and do not carry substantially different semantic content.

6. Why does the method rely on the RGB-based SpyNet for optical flow estimation? In fact, many event-based optical flow methods have already been proposed [3–5], which may be more suitable for this context.

**Reference**

[1] Tu Z, Li H, Xie W, Liu Y, Zhang S, Li B, Yuan J. Optical flow for video super-resolution: A survey. Artificial Intelligence Review. 2022 Dec;55(8):6505-46.

[2] Chan KC, Zhou S, Xu X, Loy CC. Basicvsr++: Improving video super-resolution with enhanced propagation and alignment. InProceedings of the IEEE/CVF conference on computer vision and pattern recognition 2022 (pp. 5972-5981).

[3] Wan Z, Dai Y, Mao Y. Learning dense and continuous optical flow from an event camera. IEEE Transactions on Image Processing. 2022 Nov 14;31:7237-51.

[4] Gallego G, Rebecq H, Scaramuzza D. A unifying contrast maximization framework for event cameras, with applications to motion, depth, and optical flow estimation. InProceedings of the IEEE conference on computer vision and pattern recognition 2018 (pp. 3867-3876).

[5] Gehrig M, Millhäusler M, Gehrig D, Scaramuzza D. E-raft: Dense optical flow from event cameras. In2021 International Conference on 3D Vision (3DV) 2021 Dec 1 (pp. 197-206). IEEE.

**Questions:**

See weaknesses

---

### Official Review · Reviewer_MHVF · 2025-10-28

**Soundness:** 2
**Presentation:** 2
**Contribution:** 3
**Rating:** 4
**Confidence:** 4

**Summary:**

The paper tackles event stream super-resolution by leveraging motion cues and polarity-aware modeling. It proposes MPS-PI Net, a framework that treats event frames as the main input while using positive and negative event streams as auxiliaries. A Motion-Guided Semantic Prior (MGSP) module aligns and injects semantic cues from previously reconstructed frames via optical flow to support the current frame, aiming to improve temporal coherence and reduce reconstruction difficulty. A Bipolar Semantic Interaction and Fusion (B-SIF) module first strengthens intra-polarity representations through parallel spatial-channel attention, then performs cross-polarity interaction to exploit complementarities without early fusion interference. The approach positions itself as a principled way to preserve event-specific dynamics, enhance spatio-temporal consistency, and provide a practical ESR pipeline validated on both synthetic and real data.

**Strengths:**

1.	This paper identifies two key issues in event-stream ESR: insufficient spatial–motion semantic consistency and underutilization of the complementary patterns between positive and negative polarities.
2.	This paper uses optical flow to align the historical hidden state and injects semantic cues from the previously reconstructed frame into the current reconstruction, simultaneously reducing current reconstruction difficulty and enhancing cross-frame semantic consistency.

**Weaknesses:**

1.	The effectiveness of B-SIF is mainly supported by empirical results, but lacks deeper analysis of cross-polarity interaction. The proposed cross-attention appears to be a plug-and-play Transformer application; the paper should further articulate the motivation for the design. Why is cross-polarity interaction reasonable? Why is it effective?
2.	The framework relies on optical flow to align the previous hidden state. Under large motion, non-rigid deformation, occlusion, or flow estimation errors, could error propagation introduce significant negative effects? This requires further discussion.
3.	What concrete benefits does Event Stream SR bring to event-based vision? A deeper discussion is recommended; otherwise, the motivation of the paper may appear unclear.
4.	The comparative methods in Table 4 should include citations.

**Questions:**

Please see the weaknesses part.

---

### Official Review · Reviewer_vRZF · 2025-10-31

**Soundness:** 3
**Presentation:** 2
**Contribution:** 2
**Rating:** 4
**Confidence:** 3

**Summary:**

The paper introduces a Motion-Guided Prior Support and Polarity Interaction Network for event stream super-resolution.
 Reported results on synthetic and real datasets show quantitative gains over selected baselines.

**Strengths:**

Leverages inter-frame motion and polarity-specific streams to enhance ESR features.

Clear modular design with ablations indicating component contributions.

Evaluations provided on both synthetic and real event datasets.

**Weaknesses:**

1 Mis-specified motivation.

Modern event sensors (e.g., Prophesee Gen4 at 1280×720; CeleX-V) already achieve high spatial resolution, rendering "low resolution" an unconvincing driver for ESR. The paper should (i) delineate specific ESR application scenarios and (ii) rigorously justify its advantages over higher-resolution sensors or alternative preprocessing techniques.

2 Llimited novelty.

The assertion that prior ESR methods merge polarities and discard information is incorrect; several works explicitly preserve polarity separation (e.g., Li et al., ICCV 2021; BMCNet, CVPR 2024). The proposed B-SIF thus appears incremental—demonstrate quantifiable benefits under controlled, matched comparisons to establish novelty.

3. Gaps in metrics and task validation.

While prior ESR literature commonly reports MSE, this work uses only RMSE, hindering direct benchmarking. More critically, validate utility on downstream tasks (e.g., event-based video reconstruction, tracking, optical flow, or detection) using standard pipelines and established metrics.

4. Limited contribution relative to added complexity.

The approach integrates existing motion-guided priors with polarity-separated attention. Reported gains are modest compared to increased complexity and latency; strengthen evidence through cross-dataset generalization, robustness to sensor noise and illumination variations, and ablation studies on flow quality, polarity imbalance, and noise levels.

**Questions:**

None

---

### Note · Authors · 2025-11-12

I have read and agree with the venue's withdrawal policy on behalf of myself and my co-authors.